# Metabolic Changes and Increased Levels of Bioactive Compounds in White Radish (*Raphanus sativus* L. cv. 01) Sprouts Elicited by Oligochitosan

**Apidet Rakpenthai [1], Gholamreza Khaksar [1], Meike Burow [2], Carl Erik Olsen [3] and Supaart Sirikantaramas [1,4,\*]**

[1] Department of Biochemistry, Faculty of Science, Chulalongkorn University, 254 Phayathai Road, Pathumwan, Bangkok 10330, Thailand

[2] DynaMo Center, Department of Plant and Environmental Sciences, University of Copenhagen, Thorvaldsensvej 40, 1871 Frederiksberg C, Denmark

[3] Section for Plant Biochemistry, Department of Plant and Environmental Sciences, University of Copenhagen, Thorvaldsensvej 40, 1871 Frederiksberg C, Denmark

[4] Natural Product Biotechnology Research Unit, Chulalongkorn University, 254 Phayathai Road, Pathumwan, Bangkok 10330, Thailand

\* Correspondence: supaart.s@chula.ac.th; Tel.: +66-2-218-5425; Fax: +66-2-218-5418

**Abstract:** The effect of oligochitosan O-80 treatment on metabolic changes in white radish (*Raphanus sativus* L.) sprouts (WRS) was investigated for its potential to enhance bioactive compounds accumulation. The seeds were germinated in deionized water containing different concentrations of oligochitosan O-80 (0 (control), 5, 10, 20 and 40 ppm). Seven-day old sprouts were harvested for metabolome analysis using liquid chromatography-mass spectrometry (LC-MS) and high-performance liquid chromatography (HPLC) for phenolic compounds and glucosinolate analysis, respectively, and spectrophotometric assays to determine the total phenolic content and antioxidant capacity. Partial least squares discriminant analysis was adopted to model the data from the LC-MS and revealed that O-80 at all tested concentrations affected the metabolite profiles of the treated WRS samples. The UV chromatogram at 320 nm showed increased levels of most sinapoyl derivatives, consistent with the increased total phenolic contents. Interestingly, glucoraphasatin (4-methylthio-3-butenyl glucosinolate), a major glucosinolate detected by HPLC, was increased by 40% in the sprouts treated with 10 ppm O-80. Our results provide compelling evidence regarding the exogenous application of oligochitosan O-80 as an elicitor of bioactive metabolites in WRS.

**Keywords:** bioactive metabolite; glucosinolate; metabolomics; oligochitosan; total phenolic content; white radish sprout

## 1. Introduction

Plants within the family, Brassicaceae, include some of the most popularly-consumed vegetables and are rich in several bioactive compounds capable of promoting human health, including carotenoids, vitamins, flavonoids and glucosinolates (GLs). The consumption of cruciferous vegetables (Brassicaceae family) has been reported to decrease the risk of developing cancer [1]. White radish sprouts (WRS; *Raphanus sativus*, L.), a member of the Brassicaceae family, is studied in the present work as a source of health-promoting bioactive compounds, such as folic acid, dietary fiber, flavonoids, polyphenolics, vitamin A, vitamin C, and GLs [2,3].

Dietary consumption of Japanese radish sprouts was reported to elicit a hypoglycemic activity in both normal and streptozotocin-induced diabetic rats, and also had the potential to alleviate

hyperglycemia in diabetic rats [4]. Previous reports have strongly suggested that the early developmental stage of edible sprouts has a higher level of bioactive compounds than the mature field-grown stage, such as broccoli and cauliflower sprouts, which contain 10–100 times higher levels of glucoraphanin (an anticarcinogen bioactive compound) than the corresponding mature plants [5,6].

A comparative study of the phytochemical composition between the sprouts and taproot or radish revealed significantly higher levels of GLs, isothiocyanates and total phenolic contents in the sprouts than in the taproots [7]. In addition, there are several reports on the beneficial activities of 4-methylthio-3-butenyl isothiocyanate (MTBITC), also called raphasatin, which is the hydrolysis product of the glucosinolate glucoraphasatin, and is predominant in radish sprouts [3,8,9]. Moreover, 4-methylsulfinyl-3-butenyl isothiocyanate, also called sulforaphene, the isothiocyanate derived from glucoraphenin, is also predominant in radish sprouts. These results provide convincing evidence that WRS have a wide range of abundant metabolites that are of great nutritional value and health benefits.

Chitosan, derived from deacetylation of chitin, is an elicitor of plant defense responses. Chitosan stimulates the formation of physiological defense responses against invading pathogens leading to the synthesis of a number of bioactive compounds, such as phytoalexins. Chitosan treatment of wheat seeds stimulated the synthesis of phenolic acids in the primary leaves, and levels of these phenolic acids along with the lignin content were enhanced significantly with increasing chitosan concentrations [10]. The potential of chitosan to promote the activity of phenylalanine ammonia-lyase (PAL) and to boost the amounts of total polyphenols in banana and strawberry has been reported [11]. Interestingly, the application of oligomeric chitosan with an 80% degree of deacetylation (O-80) and molecular mass of 45 kilo dalton (kDa) in orchid significantly enhanced the fresh weight of plantlets, inflorescence and number of flowers per inflorescence. Moreover, O-80 treatment enriched the levels of vitamins, nutrients and bioactive compounds in the orchid [12]. These characteristics of O-80 prompted us to use it as an elicitor for us in-planta study.

There have been a few studies on oligochitosan elicitation in members of the Brassicaceae family, such as the study by Barrientos Carvacho et al. [13]. In this study, a significant up-accumulation of the phenolic content in broccoli sprouts treated with oligochitosan was observed. However, to the best of our knowledge, there are no published studies in the literature regarding oligochitosan-treated WRS. Therefore, this study aimed to assess the effect of O-80 treatment on the metabolic profile of WRS in relation to the accumulation of bioactive compounds, including total phenolic content and antioxidant capacity by combining non-targeted and targeted metabolite analyses. Moreover, we germinated the white radish seeds in water containing the O-80 as its source. This is different from almost all previous studies that have exogenously applied the elicitor with a foliar spray. Our approach seems to be easier, less time-consuming and less labor intensive for further application compared to foliar application.

## 2. Materials and Methods

### 2.1. Chemicals

All reagents and internal standards used in this study were obtained from Sigma-Aldrich Inc. St. Louis, MO, USA. Organic solvents for liquid chromatography-mass spectrometry (LC-MS) and high-performance liquid chromatography (HPLC) were obtained from Merck KGaA, Darmstadt, Germany.

### 2.2. Oligochitosan (O-80) Preparation

Preparation of oligochitosan O-80 from crab shell was performed in the Carbohydrate Biotechnology Laboratory, Chulalongkorn University following the standard protocol described in Limpanavech et al. [12] and was defined as the solution containing chitosan molecules having the average molecular mass of 45,000 Da with the degree of deacetylation (DD) > 80%.

*2.3. Plant Material and Sample Preparation*

White radish (*Raphanus sativus* L. cv. 01) seeds (a popular variety in Thai markets) were purchased from Chia Tai Co., Ltd. (Bangkok, Thailand) and germinated on perlite supplied with pH-adjusted deionized water containing O-80 at 0 (control), 5, 10, 20 and 40 ppm. For each concentration, three replicates were used (for each replicate, three seeds were used). After germination the seedlings were grown in the deionized water containing the respective O-80 concentration at 25 ± 2 °C and a relative humidity of 70% for 7 d, whereupon the edible part (stem and leaf) of the WRS were collected, ground in liquid nitrogen, and stored at −80 °C until used.

*2.4. Total Phenolic Content Assay*

One mL of absolute ethanol was mixed with 100 mg of the ground WRS to extract the phenolic and antioxidant compounds following the method described in Thaipong et al. [14]. The mixture was sonicated for 10 min and then centrifuged using Eppendorf™ 5424R Microcentrifuge (Fisher Scientific Company, Toronto, ON, Canada) at 12,000 rpm (13,400× *g*) for 10 min. The supernatant was then harvested and used in determination of the total phenolic and antioxidant contents. The total phenolic content was estimated by the Folin-Ciocalteu assay according to the method described in Singleton et al. [15] using gallic acid as the reference standard. Samples were prepared by mixing a 0.5 mL aliquot of the ethanolic extract with 0.5 mL of 90% (*v/v*) ethanol, 2.5 mL of distilled water and 0.25 mL of Folin-Ciocalteu's reagent (50%). The mixture was incubated at room temperature (27 °C) for 5 min after which 0.5 mL of 5% (*w/v*) sodium carbonate was added, mixed thoroughly and incubated for 1 h in dark before the absorbance was measured at 725 nm ($A_{725}$). The total phenolic content was expressed as μg gallic acid equivalent (GAE) per mg fresh weight.

*2.5. Antioxidant Activity Assays*

2.5.1. Ferric Ion Reducing Antioxidant Power (FRAP) Assay

The FRAP assay was performed following the method described by Benzie and Strain [16] with some modifications according to Thaipong et al. [14]. The WRS ethanolic extract (150 μL) was allowed to react with 2850 μL of the freshly prepared FRAP solution for 30 min in the dark and then the concentration of formed ferrous tripyridyltriazine complex was monitored by measuring the absorbance at 593 nm ($A_{593}$) and comparing it to the standard ($A_{593}$) curve of Trolox to express the results in μg Trolox equivalent (TE) per mg fresh weight.

2.5.2. (2′-Azino-bis(3-ethylbenzothiazoline-6-sulphonic acid) (ABTS) Radical Scavenging Activity

The ABTS assay was performed according to a previously published method [17] with brief modifications according to Thaipong et al. [14]. The extracted WRS sample (150 μL) was added to 2850 μL of the prepared $ABTS^+$ solution and kept in the dark for 2 h prior to measuring the absorbance at 734 nm ($A_{734}$). The percentage decrease in the $A_{734}$ value was calculated as shown:

$$ABTS^+ \text{ scavenging effect (\%)} = ((AB - AA)/AB) \times 100$$

where AB is the $A_{734}$ of the blank sample ($t = 0$ min) and AA is the $A_{734}$ of the extract solution.

2.5.3. 2,2-Diphenyl-1-picrylhydrazyl (DPPH) Radical Scavenging Activity

The DPPH assay was performed according to the previously described method [18] with some modifications according to Thaipong et al. [14]. The WRS extract (150 μL) was added to 2850 μL of the freshly prepared DPPH solution and incubated at room temperature (27 °C) for 24 h in the dark prior to measuring the absorbance of the reaction mixture at 515 nm ($A_{515}$). The $A_{515}$ of the DPPH

radical without any antioxidant was also measured and the DPPH radical scavenging activity was then calculated as follows:

$$\text{DPPH radical scavenging activity (\%)} = ((A_0 - (A_1 - A_2))/A_0) \times 100$$

where $A_0$ is the $A_{515}$ of DPPH without crude extract and $A_1$ and $A_2$ are the $A_{515}$ of the crude extract with and without DPPH, respectively.

### 2.6. Metabolomics Study Using Liquid Chromatography-Mass Spectrometry (LC-MS) Analysis

In order to prepare the samples for the metabolomics analysis, 100 mg of the ground WRS from the respective treatment was extracted with a 2:5:2 (*v/v/v*) of dichloromethane/methanol/water [19]. Before the extraction, a known amount of amygdalin was added as an internal standard. The extracted metabolites were then separated into polar and non-polar phases. The polar extracts were harvested, dried under vacuum and the obtained residues were resuspended in 100 µL of 10% (*v/v*) methanol. LC-MS analysis was performed using a LC (Agilent 1200 SL series, Agilent Technologies, Germany) coupled to a Bruker micrOTOF-Q mass spectrometer (Bruker Daltonics, Bremen, Germany). An Xbridge C18 column (Waters, Milford, MA; 3.5 µm, 2.1 × 150 mm) was used at a flow rate of 0.2 mL min$^{-1}$. Solvent A was 0.1% (*v/v*) formic acid, and solvent B was acetonitrile with 0.1% (*v/v*) formic acid. The cycle time for the method was 30 min using the following gradient: 0–1 min, isocratic 10% B; 1–18 min, linear gradient 10–35% B; 18–20 min, linear gradient 35–100% B; 20–23.5 min, isocratic 100% B; 23.5–30 min equilibration to 10%B. The spectrometer was run in electrospray mode, observing positive ions over the *m/z* range 50–1000. The spectra were calibrated using sodium formate clusters generated by temporarily switching the spectrometer inlet to a stream of 2 mM sodium formate in 2-propanol during the first and last 1, and 5 min of each LC-MS run. UV-VIS spectra were recorded in parallel.

### 2.7. Data Processing Using Chemometric Tools

The LC-MS data were exported as a comma-separated value (*.csv) file format and aligned using metAlign [20], where data (pre-) processing included deriving peak-picked data from the raw data, automated baseline correction and subsequent spectral alignment. To investigate the differences among metabolites, partial least squares discriminant analysis (PLS-DA) was used to cluster and remove outliers among samples. PLS-DA was generated using Metabo Analyst 4.0, an open source R-based program [21]. For the PLS-DA, the values were sum normalized and log$^2$ transformed. Hierarchical clustering analysis (HCA) combined with heat map was generated using R.

### 2.8. Glucoraphasatin Content Analysis

To quantify the glucoraphasatin content of each WRS sample, 50 mg of ground WRS was extracted as mentioned in 2.5, except that prior to the extraction, 50 µg of sinalbin was added to each sample as an internal standard. The vacuum-dried extracts were redissolved in 150 µL of 10% (*v/v*) methanol for 5 min at room temperature, clarified by centrifugation at 2500× *g* for 10 min, and the supernatant was harvested and loaded on a column containing 0.4 mL of a 10% (*w/v*) suspension of DEAE Sephadex A-25 in H$_2$O. Columns were sequentially washed with 1 mL 70% (*v/v*) methanol, 1 mL water and 1 mL 0.02 M MES buffer (pH 5.2) before 50 µL of sulfatase solution was applied and left overnight at room temperature. The desulfated GLs were then eluted with 2 × 0.8 mL of 60% (*v/v*) methanol, dried at 50 °C in a nitrogen stream, and then redissolved in 0.4 mL water. Desulfo-GLs were separated by HPLC on an Agilent HP1100 Series instrument equipped with a C-18 reversed phase column (LiChrospher RP18ec, 250 × 4.6 mm, 5 µm particle size), using the following gradient (mobile phase A, water; B, acetonitrile) at a flow rate of 1 mL min$^{-1}$ at 25 °C (injection volume 50 µL): 0–5 min, gradient from 1.5–7% B; 5–10 min, gradient from 7–25% B; 10–14 min, gradient from 25–80% B; 14–17 min, isocratic at 80% B; 17–19 min, gradient from 80–35% B; 19–21 min, gradient from 35–1.5% B; 21–24 min,

isocratic at 1.5% B. Desulfo-GLs were identified based on comparison of the retention times and UV absorption spectra with those of known standards and quantified based on $A_{229}$ using published response factors [22].

*2.9. Statistical Analysis*

Each assay was performed with three biological replicates and the data are presented as the mean ± SEM (standard error of mean). Statistical analysis of the data was conducted by analysis of variance (ANOVA) and Duncan's multiple range tests (DMRT) using the statistical package for social sciences (SPSS) software, version 20.

## 3. Results and Discussion

*3.1. Non-Targeted Metabolome Analysis of WRS Treated with Oligochitosan O-80*

The PLS-DA clearly demonstrated an O-80 concentration-dependent change in the metabolite profile of WRS grown in the presence of O-80 for 7 d (Figure 1A). The individual samples within the treatment group (represented by different colors) formed distinct clusters. The total variance of the data explained by the cumulative contribution ratio of the PLS-DA was 56.6% and was comprised of two main factors that explained 34% and 22.6% of the variance in the first principle component (PC1) and the second principle component (PC2), respectively. PC1 resolves the dose-dependent metabolic changes. The sprouts exposed to 20 and 40 ppm of O-80 had a higher PC1 value compared to the others, while the WRS exposed to the lower dose (treatment with 5 ppm of O-80) clustered closely to the control treatment (0 ppm). The PLS-DA demonstrated that the sprouts treated with O-80 showed significant changes in their metabolic profiles with WRSs under 20 and 40 ppm of O-80 harboring the most significant changes in metabolite profiles compared to the control. Notably, the PLS-DA model offers the variable in projection (VIP) scores to estimate the characteristic of highest importance in explaining the relationship among variables. Many masses with high VIP scores showed the decrease in concentrations upon the treatment (Figure 1B).

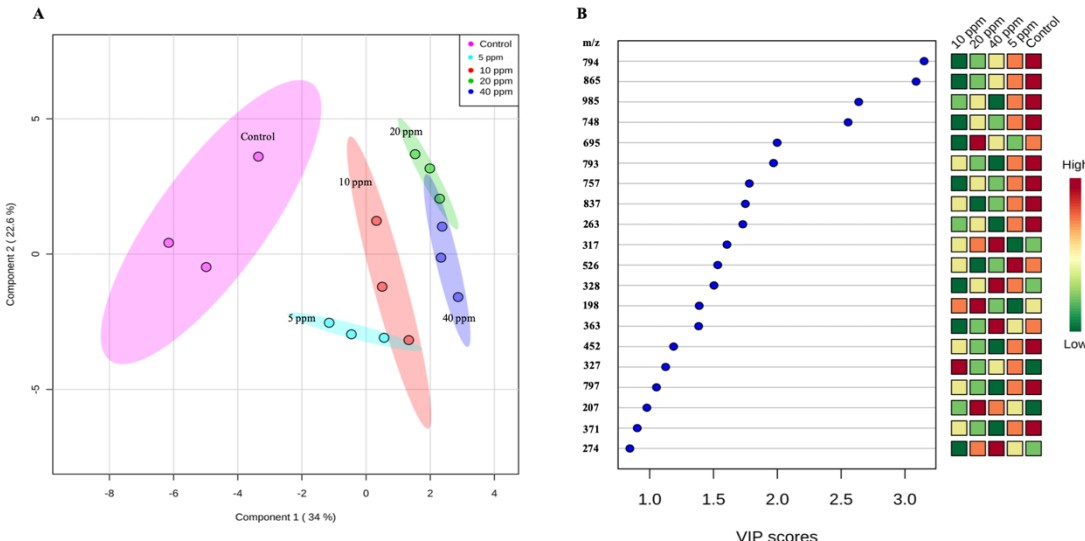

**Figure 1.** Partial least squares discriminant analysis (PLS-DA) of metabolites in oligochitosan O-80-treated white radish sprouts (WRS) samples (0 (control), 5, 10, 20 and 40 ppm) analyzed by LC-MS in the positive ion mode. Each spot represents a biological replicate (**A**). Variable in projection (VIP) scores of the top 20 metabolites (**B**).

### 3.2. Hierarchical Clustering Analysis (HCA)

The HCA of the metabolite profiles revealed two groups that were separated from each other based on the dosage of applied O-80, with high concentrations (20 and 40 ppm) of applied O-80 forming one group and low concentrations (5 and 10 ppm) the other (Figure 2). The clustering of all significantly changed metabolite levels formed six different patterns. However, the current database used to annotate the metabolites has limitations, with some obtained masses having insufficient information to the identity the metabolite. Consistent with this, many previously published studies have raised concerns regarding mass spectra due to limited available data for detecting their secondary metabolites [2,23].

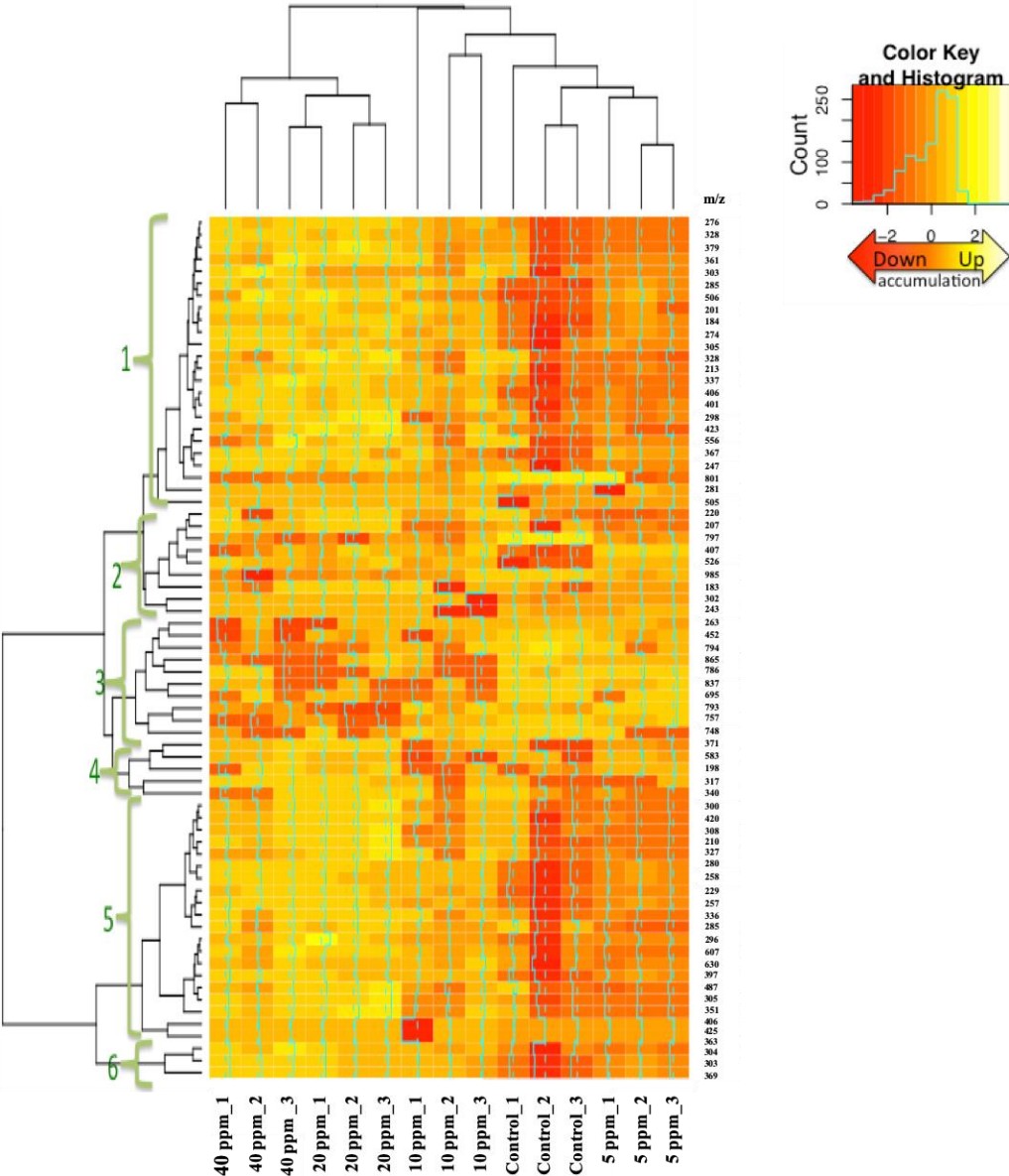

**Figure 2.** Hierarchical clustering analysis (HCA) of the metabolite levels (represented as mass-to-charge ratio (*m/z*) signal intensities) affected by different doses of oligochitosan (O-80) treatment. The heat map shows the relative metabolite abundance for each replicate.

There was a significant change in metabolite levels of oligochitosan-treated WRS compared to the control. The increased alteration in the metabolites fold-changes was consistent with increasing O-80 concentrations (Table 1). The highest fold-change was seen for the compound(s) with a $[M + H]^+$ of

317 *m/z* (cluster 4), which was up-accumulated by 9.89 ± 0.16 folds under 40 ppm of O-80 treatment compared to the control. Moreover, the compound(s) with a mass of 748 (cluster 3) and 985 *m/z* (cluster 2) showed a down-accumulation compared to the control. These metabolites with significant fold changes were among those with a VIP score of more than one. Taken together, the PLS-DA analysis and heat map showed the significant changes in the levels of some important metabolites under O-80 treatment. The significant changes in metabolic profiles of treated WRS under oligochitosan treatment, specifically at high concentrations (20 and 40 ppm), provide convincing evidence regarding the role of oligochitosan O-80 treatment in altering metabolic levels of WRS. Notably, due to the insufficient data available for accurate matching, these compounds remain unknown for us. However, we focused on these metabolites since they harbor a significantly higher fold change in treated WRS compared to the control. These unknown metabolites could be the subject of further investigations, specially with regard to their potential human health benefits.

**Table 1.** List of the ten highest and most significantly changed *m/z* signal intensities.

| Mass-To-Charge (*m/z*) Ratio | Retention Time (min) | HCA * | Fold Change ** (Mean ± Standard Error of Mean (SEM), *n* = 3) at Oligochitosan Concentration of: | | | |
|---|---|---|---|---|---|---|
| | | | 5 ppm | 10 ppm | 20 ppm | 40 ppm |
| 274 | 11.8 | 1 | 2.29 ± 0.23 [d] | 3.54 ± 0.24 [c] | 4.41 ± 0.10 [b] | 5.06 ± 0.06 [a] |
| 305 | 18.0 | 1 | 1.82 ± 0.18 [d] | 3.60 ± 0.28 [c] | 4.50 ± 0.36 [b] | 5.45 ± 0.22 [a] |
| 184 | 7.4 | 1 | 2.21 ± 0.21 [c] | 3.42 ± 0.34 [b] | 4.42 ± 0.09 [a] | 3.88 ± 0.09 [ab] |
| 243 | 12.9 | 2 | 1.67 ± 0.05 [c] | 1.80 ± 0.90 [bc] | 3.35 ± 0.04 [ab] | 4.12 ± 0.36 [a] |
| 985 | 8.0 | 2 | 0.49 ± 0.10 [a] | 0.47 ± 0.14 [a] | 0.22 ± 0.04 [ab] | 0.13 ± 0.06 [b] |
| 748 | 1.9 | 3 | 0.59 ± 0.12 [a] | 0.45 ± 0.12 [a] | 0.03 ± 0.02 [b] | 0.08 ± 0.07 [b] |
| 317 | 13.0 | 4 | 0.55 ± 0.45 [b] | 2.71 ± 1.60 [b] | 7.40 ± 0.64 [a] | 9.89 ± 0.16 [a] |
| 198 | 3.1 | 4 | 0.80 ± 0.17 [c] | 1.23 ± 0.80 [c] | 6.99 ± 0.76 [a] | 4.71 ± 0.20 [b] |
| 257 | 11.8 | 5 | 2.02 ± 0.18 [c] | 3.01 ± 0.10 [b] | 3.68 ± 0.26 [a] | 4.01 ± 0.08 [a] |
| 285 | 11.8 | 5 | 1.87 ± 0.15 [d] | 2.65 ± 0.17 [c] | 3.52 ± 0.09 [b] | 4.01 ± 0.11 [a] |

\* HCA cluster number. ** Fold-change is relative to the control (0 ppm). Means within each column followed by a different lower-case letter are significantly different ($p < 0.05$; DMRT).

### 3.3. Identification of Sinapoyl Derivatives by LC-DAD-MS Chromatography

According to previous studies on sinapic acids [24,25], several of the most commonly available phenolic compounds in WRS are classified into sinapic acids, especially when compared to others in the sinapoyl ester category, such as sugar esters (glycoside) or as esters combined with organic compounds. The unique MS spectrum of these derivatives is at *m/z* 207 [M + H]$^+$ relying on the 'ReSpect for phytochemicals' database. Furthermore, the universal property of members this compound class is their ability to act as free radical scavengers, neutralizing superoxide anion, hydroxyl radicals and nonradical oxidants like hypochlorite [24].

As shown in Table 2, the increased amount of sinapoyl derivatives was positively correlated with the O-80 treatment at all doses compared to the control and was dose-dependent up to 20 ppm except for molecular ion *m/z* of 358. The largest positive fold changes were observed for peak number 2 at 20 ppm O-80 and peak number 5 at 10 ppm O-80, whilst peak number 4 was down-accumulated with increasing O-80 concentrations to a minimum with 20 ppm O-80. As for the metabolite annotation, there were insufficient matches to the database of MS/MS fragments, KNAPSACK, RIKEN (ReSpect), making absolute quantification of sinapoyl derivates impossible. From the examined LC-DAD-MS profile, peak number 3 was elucidated to be sinapic acid conjugate, which has been suggested to be involved in plant defense responses after fungal infection [26]. However, the other four peaks have not been annotated yet as insufficient data exists for accurate matching to the database. Sinapoyl derivatives are derived from the phenylpropanoid pathway, which plays a role in biotic stress resistance against fungal attack [26,27]. In addition, these compounds have beneficial human health characteristics, such as antioxidant, anti-inflammatory, anticancer and anti-anxiety activities.

**Table 2.** Changes in sinapoyl derivative levels treated with various oligochitosan O-80 concentrations.

| Peak No. | Retention Time (min) | Molecular Ion [M + H]$^+$ (*m/z*) | Fragment [M + H]$^+$ (*m/z*) | Proposed Bioactive Compound | Fold Change * (Mean ± SEM, *n* = 3) at Oligochitosan Concentration of: | | | |
|---|---|---|---|---|---|---|---|---|
| | | | | | 5 ppm | 10 ppm | 20 ppm | 40 ppm |
| 1 | 6.9 | 487 | 207, 369, 487 | Sinapic acid conjugate | 1.17 ± 0.05 [a] | 1.18 ± 0.07 [a] | 1.35 ± 0.05 [b] | 1.34 ± 0.09 [a] |
| 2 | 13.0 | 363 | 207, 363 | Sinapoyl malate | 1.11 ± 0.08 [a] | 1.27 ± 0.14 [a] | 1.55 ± 0.02 [a] | 1.26 ± 0.06 [a] |
| 3 | 14.4 | 777 | 207, 369 | Sinapic acid conjugate | 0.94 ± 0.05 [b] | 1.09 ± 0.11 [a] | 1.31 ± 0.08 [b] | 1.26 ± 0.07 [a] |
| 4 | 15.4 | 358 | 207, 336 | Sinapic acid conjugate | 1.09 ± 0.38 [ab] | 0.82 ± 0.49 [a] | 0.21 ± 0.03 [c] | 0.44 ± 0.15 [b] |
| 5 | 16.9 | 615 | 207, 351, 369 | Sinapic acid conjugate | 0.82 ± 0.20 [b] | 1.42 ± 0.42 [a] | 1.35 ± 0.15 [b] | 1.12 ± 0.09 [a] |

* Fold-change is relative to the control (0 ppm). Means within each column followed by a different lower-case letter are significantly different ($p < 0.05$; DMRT).

### 3.4. Antioxidant Content

At high O-80 concentrations (20 and 40 ppm), the treatment negatively affected the antioxidant machinery, as indicated by the reduced FRAP values, whereas at 5 and 10 ppm oligochitosan O-80 the FRAP values were not significantly different from the control (Figure 3A). For the ABTS radical scavenging activity, the O-80 treatment at 5 ppm slightly enhanced the % ABTS activity compared to the control, whereas the % ABTS activity values in the other O-80 treatments were not significantly different to the control (Figure 3B). Finally, the DPPH radical scavenging activity was not significantly affected by the O-80 treatment at any tested concentration (Figure 3C). Thus, the effect of O-80 treatment on the antioxidant machinery of WRS remains unclear. The different methods of evaluating the antioxidant activity of O-80-treated sprouts yielded different results, perhaps reflecting that simultaneously present antioxidants in plant extracts show different mechanisms of action. Antioxidants react with free radicals by different mechanisms; hydrogen atom transfer (HAT) or single electron transfer (SET) mechanism; or the combination of both HAT and SET mechanisms. During the HAT mechanism, the free radical takes one hydrogen atom from the antioxidant, and the antioxidant itself becomes a radical. However, in the SET mechanism, the antioxidant provides an electron to the free radical and itself then becomes a radical cation. DPPH and FRAP follow the SET mechanism while ABTS follow the HAT mechanism [28]. This difference in the mechanism of action could clearly be observed in our results where FRAP and DPPH showed a quite similar trend which was different from that of ABTS. Another point to be noted is the high values of standard deviations in the FRAP result. We currently do not know the reason, but the results are calculated and represented accurately.

### 3.5. Total Phenolic Content

The total phenolic content of the WRS was significantly increased (2.8-fold) by treatment with O-80 at 5 ppm (optimum concentration). However, increasing the O-80 concentration resulted in a less marked increase in the total phenol level (Figure 4). These results are similar to those reported by Barrientos Carvacho et al. [13], who reported a significant up-accumulation of the phenolic content in broccoli sprouts treated with oligochitosan at 10 μM compared to the control, while at higher oligochitosan concentrations (50 and 90 μM), the phenolic content of the broccoli sprouts declined. Exogenously applied elicitors, such as oligochitosan, have been found to significantly induce the accumulation of phenolic compounds in many varieties of Brassica by stimulating phenylpropanoid pathways [13]. Consistently, our results showed that the exogenous application of oligochitosan significantly enhanced the total phenolic contents in WRS.

### 3.6. Effect of Oligochitosan O-80 Treatment on the Glucoraphasatin Content

Glucoraphasatin is the major GL in WRSs and accounts for more than 93% of the total GL contents [29]. The glucoraphasatin content of WRS was increased by treatment with O-80, increasing as the O-80 dose increased to 10 ppm (optimum concentration) and remained stable at this elevated level as the O-80 dose further increased to 40 ppm (Figure 5). Notably, the level of MTBITC, isothiocyanate produced from glucoraphasatin and thus the predominant isothiocyanatein WRSs, was significantly decreased in WRS under methyl jasmonate treatment [23].

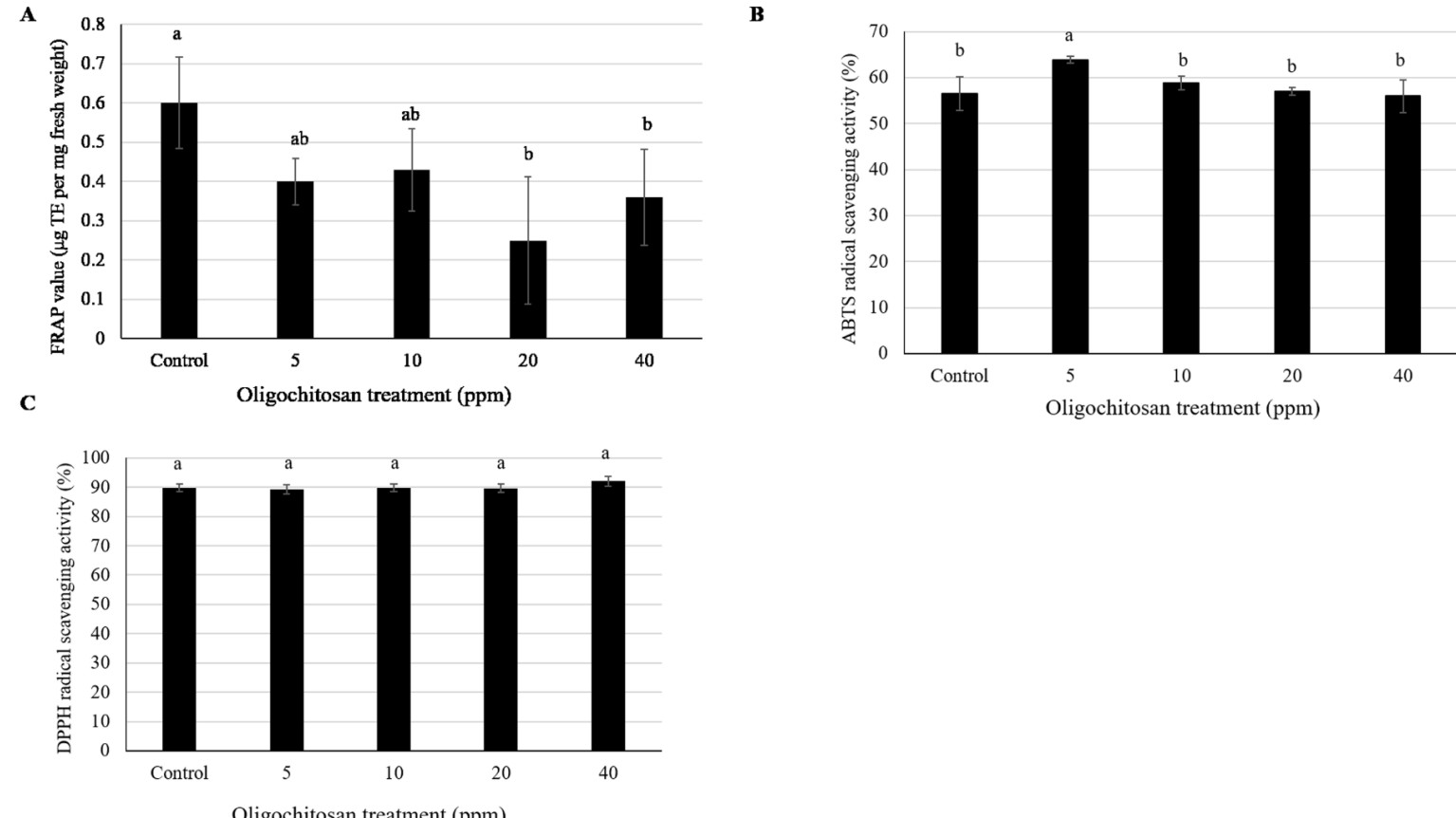

**Figure 3.** Antioxidant capacity in WRS exposed to oligochitosan O-80, as measured by the (**A**) Ferric ion reducing antioxidant (FRAP), (**B**) 2′-azino-bis (3-ethylbenzothiazoline-6-sulphonic acid (ABTS) and (**C**) 2,2-diphenyl-1-picrylhydrazyl (DPPH) assays. For each assay, bars with different letters above them were significantly different according to Duncan's multiple range test ($p < 0.05$). Control is O-80 at 0 ppm.

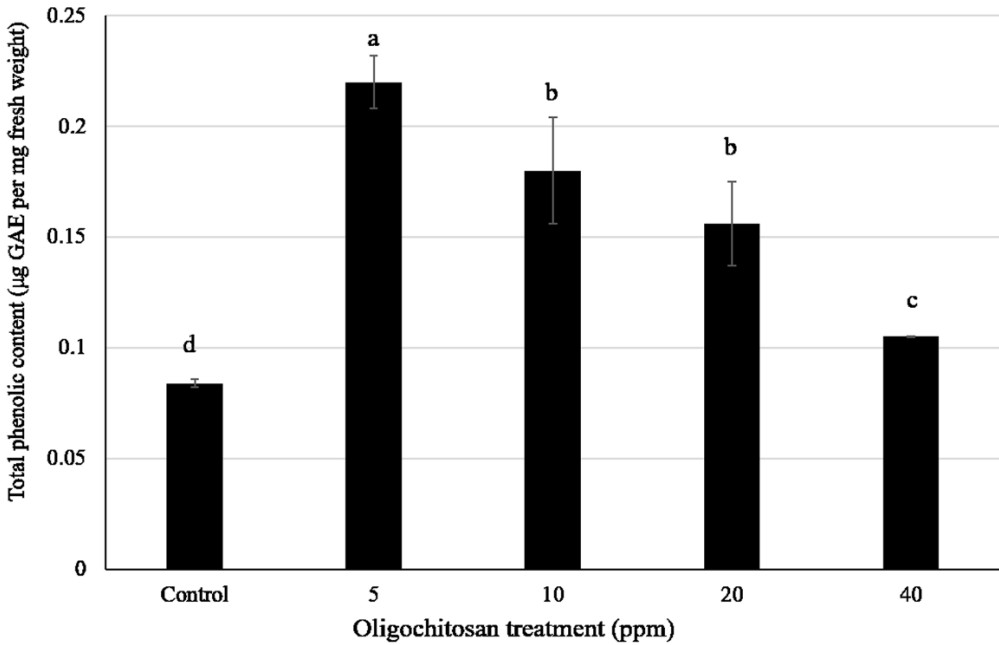

**Figure 4.** Total phenolic content in WRS extracts treated by oligochitosan O-80. Bars with different letters above them were significantly different according to Duncan's multiple range test (*p* < 0.05). Control is O-80 at 0 ppm.

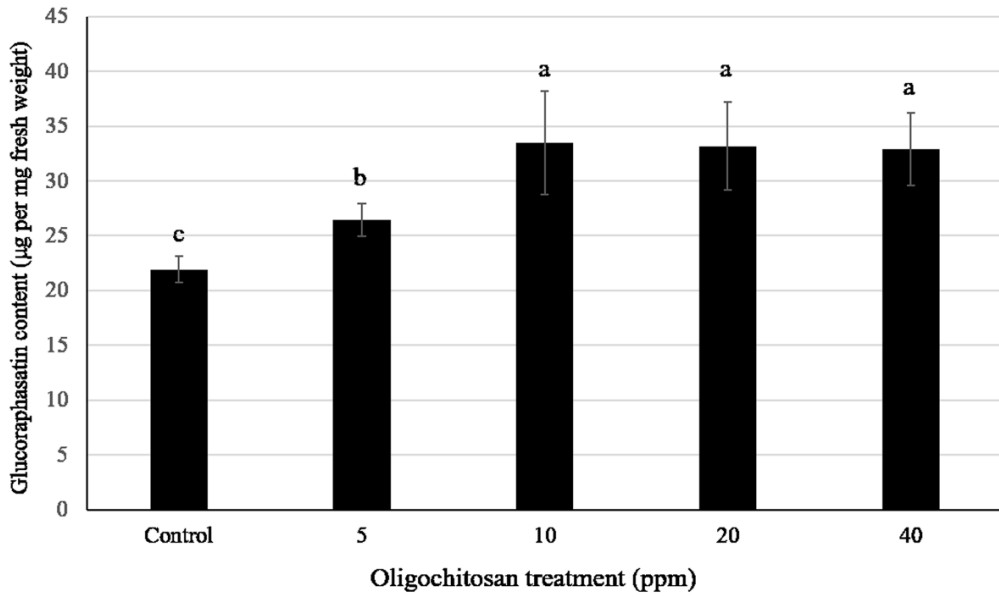

**Figure 5.** Glucoraphasatin content in WRS treated by oligochitosan O-80. Bars with different letters above them were significantly different according to Duncan's multiple range test (*p* < 0.05). Control is O-80 at 0 ppm.

It should be noted that the increase in the glucoraphasatin content after O-80 treatment followed a pattern opposite to that of the total phenolic content. The total phenolic content was maximal with O-80 at 5 ppm and decreased with higher O-80 levels, while glucoraphasatin was maximal with O-80 at 10 ppm and did not decrease with higher O-80 concentrations.

This reflects the different biosynthetic pathways of these compounds and their precursors. Phenolic compounds are synthesized through the phenylpropanoid pathway from phenylalanine, while glucosinolate is synthesized from methionine through a different pathway.

There are several reports in the literature indicating that chitosan, recognized by chitin receptors, explicitly induces defense responses involved in plant innate immunity, including GLs biosynthesis [30,31]. The potential role of increased GLs levels in plants is correlated with biotic stress tolerance due to the biocontrol characteristics of GLs against herbivores and pathogens. The increased levels of glucoraphasatin following O-80 treatment seen here concur with the enhanced levels of aliphatic GL in oligochitosan-treated broccoli sprouts [32].

Taken together, exogenous application of O-80 significantly altered the metabolic profiles of WRS compared to the control. These significant differences in the metabolite concentrations were clearly observed in the total phenolic and glucoraphasatin contents of treated sprouts compared to the control. Notably, as closely separated by PLS-DA, the metabolite profiles of the treated sprouts under high concentrations of O-80 (20 and 40 ppm) were similar.

## 4. Conclusions

Exogenously-applied O-80 oligochitosan significantly enhanced the GL and phenolic contents, as well as untargeted bioactive compounds, in WRS. The significant shift in the metabolic profiles of WRS under high concentrations of O-80 (20 and 40 ppm) highlights the role of oligochitosan O-80 as an elicitor of health-beneficial secondary metabolites in WRS. Some of the unknown metabolites that displayed significantly altered levels after O-80 treatment could be the subject of further investigation to evaluate potential human health benefits.

**Author Contributions:** Conceived the research and designed the experiments, S.S.; performed most of the experiments and drafted the manuscript, A.R.; performed the glucosinolate content analysis, M.B.; performed the LC-MS analysis, C.E.O.; analysed the results, interpreted the data and edited the manuscript, S.S. and G.K. All authors have read and approved the final manuscript.

**Funding:** This research was funded by Chulalongkorn University [GRU 6101133003-1] (to S.S), Ratchadapisek Somphot Fund for Postdoctoral Fellowship, Chulalongkorn University (to G.K.) and Danish National Research Foundation (DNRF) [Grant 99] (to M.B).

**Acknowledgments:** We express our sincere gratitude to Cerebos (Thailand) Limited and Chulalongkorn University for financial support. We also thank Assistant Professor Rath Pichyangkura for oligochitosan preparation.

**Conflicts of Interest:** The authors declare no conflict of interest.

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
