# Peer review of "Metabolic Changes and Increased Levels of Bioactive Compounds in White Radish (Raphanus sativus L. cv. 01) Sprouts Elicited by Oligochitosan"

_agronomy, doi:10.3390/agronomy9080467_

Round 1

Reviewer 1 Report

The manuscript “Metabolic changes and increased levels of bioactive compounds in Raphanus sativus (radish) sprouts elicited by oligochitosan treatment” results of great interest for the development of plant foods rich in bioactive compounds with health-promoting benefits.

The analysis of sinapic acids derivatives, total phenolic compounds and antioxidant capacities described in this work are very interesting to understand the changes in the phytochemical profile due to the treatments with oligochitosan. However, the utility of the non-target analysis of the metabolome by LC-MS is not clear, as authors did not describe the kind of compounds/metabolic pathways that are may be affected by the treatments (results shown in Figure 1 and 2). Therefore, it is necessary to improve the discussion of results in order to relate the differences in the metabolome with changes in bioactive compounds concentrations, as the title of this work is “Metabolic changes and increased levels of bioactive compounds…”

On the other hand, some corrections should be done before publication:

TITLE

I suggest the following changes:

“Metabolic changes and increased levels of bioactive compounds in white radish (Raphanus sativus cv. Chinese) sprouts elicited by oligochitosan”

ABSTRACT

-          Lines 19-21: Please rewrite this sentence: “using liquid chromatography-mass spectrometry (LC-MS) and high performance liquid chromatography (HPLC) for phenolic compounds and glucosinolates analysis, and spectrophotometric assays to determinate the total phenolic content and antioxidant capacity”.

INTRODUCTION

Do not confuse genus (Raphanus, Brassica, etc.) or family (Brassicaceae) in the cruciferous vegetables.

-          Line 33: Please rewrite this sentence: “Plants within the family Brassicaceae include some…”   Because Brassica genus is different than Raphanus genus and both are included in the Brassicaceae family.

-          Line 35: Please rewrite this sentence: “The consumption of cruciferous vegetables (Brassicaceae family) has been…”

-          Line 37: Please rewrite this sentence: “White radish sprouts (WRS; Raphanus sativus, L.), a member of the Brassicaceae family, is studied in the present work as a source of health-promoting bioactive compounds, such as…”

-          Line 46: Reference 6 (Barillari et a., 2006) should be changed by another one which shows a higher amount of bioactive compounds in sprouts compared to the mature plant. In this reference (Barillari et a., 2006) only cited other works.

-          Line 50: I recommend to explain that MTBITC, also called raphasatin, is the hydrolysis product of the glucosinolate glucoraphasatin, predominant in radish sprouts.

-          I suggest to add information (in Line 51 after references 3, 8, 9) about 4-methylsulfinyl-3-butenyl isothiocyanate, also called sulforaphene, the ITC derived from glucoraphenin, which is also predominant in radish sprouts.

-          Lline 62: Is “O-80” the abbreviation of Oligochitosan? Place it in parentheses. Why do you use this term? What’s the meaning of 80? Describe it in the introduction.

-          Line 67: Please, delete (mustard or cabbage family).

-          Line 70: Use subscript numbers. H2O2.

-          Line 75: Total phenolic compounds are bioactive compounds. Please, rewrite this sentence.

MATERIAL AND METHODS

-          Line 88: The acknowledgement to Prof. Pichyangkura should be placed in the section “Acknowledgement”. Authors can explain that the isolation of oligochitosan was performed in “name of the institute”.

-          Line 95: Which pH did you used?

-          Line 96: What was the fresh weight of 3 sprouts (1 sample)?

-          Line 102: Why did you choose ethanol absolute for the extraction? Usually ethanol or methanol with water is used.

-          Line 109: 0.5 mL

-          Line 36: Correct it:  27 °C 

-          Line 143: Do you mean with LC, HPLC (Agilent 1200)?

-          Line 168: Why did you analyse only glucoraphasatin? What about the rest of GLS?

RESULTS

-          Figure 1. I would recommend to use another color for C-0 or C-5 as when it is printed in black and white there is no difference.

-          Could authors say how many metabolites (aprox.) is the LC-MS detecting to evaluate the non-targeted metabolite profile? Is this information collected in the Figures 1? If yes, please, explain in section 3.1. that this is the non-target analysis.

-          Do you know what compounds are 317, 748 and 985 (m/z)? What is the importance of these data if the compounds are not identified?

-          The same with Table 1. What is the meaning of these increases of compounds that are no identified or related to any bioactive compound family?

-          Table 2: In my opinion, the fold change could be see also in the results of sinapic acids content per gram of sprouts. I recommend writing in the table the contents of sinapic acids instead or the fold change.  Please, add the results of control samples. In addition, the discussion of results comparing these radish sprouts contents of sinapic acids with other works is needed.

-          Please, enrich the discussion of section 3.3., as higher concentrations of compounds related to higher concentrations of the treatment are only found in molecular ion 777.

-          Figure 3: Standard deviations of FRAP results are very high, do you have any idea about why it happened?

-          Line 272: Each antioxidant capacity method involves a different mechanism of action of the antioxidants. Please, discuss it better in the section 3.4. to explain your results.

-          Line 293: Cite the work of Carvacho in the introduction instead of other work with other variety of sprouts or species of plant.

REFERENCES

Some grammar corrections should be done:

-          3:  Brassica sprouts

Reviewer 2 Report

The authors investigated the change of phytochemical levels and antioxidant activity in radish sprout treated oligochitosan O-80. This manuscript arouses interest in the chitosan as fertilizer to improve phenolic compounds and glucosinolates. The manuscript is fall within the scope of this journal, but requires some points to be clarified before publication.

In materials and methods (2.7. Data processing using chemometric tools), the authors describe that “PLS-DA was used to cluster and remove outliers among samples based upon SIMCA” (Page 4, line 165). However, the figure 1 seems to be obtained from MetaboAnayst, not SIMCA.

Please clearly reveal a point what the authors want to obtain from results of PLS-DA and heat map. The relationship with other results (sinapoyl derivatives, antioxidant activity, phenolic compounds, and glucosinolates) is also not clear.

PLS-DA offers variable importance in the projection (VIP) value. The VIP value is important to find the contributed metabolites for separation of samples. By using the results obtained from PLS-DA and heat map data together, it seems to be found meaningful metabolites.

Figure 2 (page 7) :  The number of samples in heat map and clustering tree is 71, but the number of m/z labels (labelled right of heat map) is 69.

Page 8, line 251 : The authors describe the peak number 3 is “sinapoly malate” in table 2, but peak number 3 is “sinapic acid conjugate” in table 2.

The authors focused on sinapoyl derivatives. The targeted metabolic profiling seems to be more suitable in this study.

Page 1, line 37, page 2, line 67 and 70 : Please use italic font to species name.

Page 3, line 123 and 133 : Please check the full name of ABTS and DPPH. The format (numbering) of title is incorrect.

Page 4, line 153 : The flow rate unit is incorrect to format.

Page 5, line 185 : Please confirm the format of “A229 nm”

Page 8, table 1 : The number (m/z ratio 274, RT 11.8) in second row is bold and table is underlined. Is there special meaning in this?

Reference : Some of reference are not arranged. No. 13, 18, and 19.
